# OPTIMIZING PSEUDO-BOOLEAN POLYNOMIALS USING HYPERGRAPH NEURAL NETWORKS

## ABSTRACT

The challenge of solving NP-hard combinatorial optimization problems with deep learning has attracted considerable interest. Modern graph neural networks are capable of efficiently solving problem instances in an unsupervised manner, however the graph structure limits the scope of their application. We present a novel approach, hereafter referred to as PB-HGNN, which employs unsupervised Hypergraph Neural Networks (HGNNs) to solve the polynomial unconstrained binary optimization (PUBO) problem. By representing the high-order terms of the pseudo-Boolean polynomials as hyperedges in a hypergraph, the HGNN is enable to capture intricate variable interdependencies beyond pairwise interactions. As a result, our framework provides the possibility of solving a wide range of discrete problems that have not previously been addressed by neural networks. We evaluate PB-HGNN on random higher-order pseudo-Boolean polynomials, including the Sherrington-Kirkpatrick model, and max-3-SAT instances. Our results show that PB-HGNN outperforms baselines on these problems and is capable of solving large-scale PUBO instances.

## 1 INTRODUCTION

A pseudo-Boolean function is a mapping $f : \mathbb{B}^n \to \mathbb{R}$, where $\mathbb{B} = \{0, 1\}$ is a Boolean domain. All pseudo-boolean functions can be uniquely represented as multilinear pseudo-Boolean polynomials Hammer & Rudeanu (1968); Boros & Hammer (2002) of the following form:

$$f(x_1, ..., x_n) = \sum_{S \subseteq [n]} c_S \prod_{j \in S} x_j, \ \ x \in \mathbb{B}^n. \tag{1}$$

The general problem to optimize (1) is known as Polynomial Unconstrained Binary Optimization (PUBO) problem. $deg(f) = \max_{c_S \neq 0} |S|$ is the degree of pseudo-boolean polynomial. $deg(f) = 2$ yields the well-known Quadratic Unconstrained Binary Optimization (QUBO) problem. Recently, there has been a resurgence of interest in encoding various Combinatorial Optimization (CO) problems as QUBOs Lucas (2014); Glover et al. (2022). This is largely due to the development of specialized computational devices such as quantum annealing Boixo et al. (2013) and coherent Ising machines Wang et al. (2013), which exploit the interaction between QUBO solutions and the fundamental states of physical systems of the Ising Hamiltonian Vadlamani et al. (2020). In the last few years, new heuristic algorithms based on unsupervised graph neural networks (GNNs) were proposed for solving CO problems formulated as QUBO Schuetz et al. (2022a); Pugacheva et al. (2024); Ichikawa (2024). They encode a problem's binary variables as graph nodes and its nonzero quadratic terms as edges, and then train a GNN to minimize the continuous relaxation of the QUBO objective.

In this paper, we address the higher-order PUBO problem, when $deg(f) \geq 3$. In combinatorial optimization, it has been widely applied for modeling a maximum satisfiability Christian et al. (2014), uncapacitated facility location Goldengorin et al. (2003); Beresnev (1973); Hammer (1968), and the $p$-median AlBdaiwi et al. (2009) problems. Other important applications occur in statistical mechanics Bernasconi (1987); Liers et al. (2010) , computer vision Ishikawa (2011); Fix et al. (2015), protein folding Babbush et al. (2014), etc. Classical approaches for optimizing higher-order $f$ rely on reducing the polynomial degree through quadratization methods Verma et al. (2021);

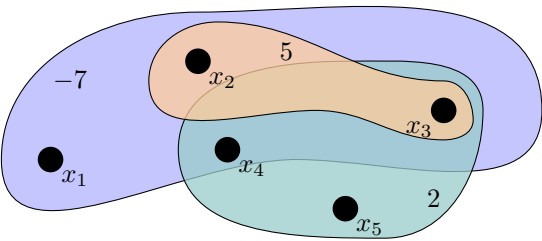

Figure 1: Hypergraph representation of $f(x_1, \ldots, x_5) = 5x_2x_3 + 2x_3x_4x_5 - 7x_1x_2x_3x_4$

Chancellor et al. (2016). By introducing additional variables, the problem can be reformulated as an integer linear program or a QUBO, which can then be solved using corresponding traditional algorithms.

While in quantum and quantum-inspired optimization it is now standard to solve PUBO *directly*—without reducing to QUBO—via higher-order Ising Hamiltonians Chermoshentsev et al. (2021); Stein et al. (2023); Kanao & Goto (2022), the neural combinatorial optimization literature still predominantly relies on explicit QUBO reductions or other objectives. In this work we introduce **PB-HGNN**, a hypergraph-based unsupervised framework that operates *natively* on higher-order pseudo-Boolean polynomials: each monomial $x^S$ becomes a hyperedge on $S$ with weight $c_S$, as shown in the fig. 1, and a lightweight residual hypergraph block produces instance-specific probabilities that are refined across epochs via an epoch-level memory. By staying in the original higher-order domain, PB-HGNN preserves multi-way interactions and simplifies the modeling pipeline.

We evaluate PB-HGNN on two representative benchmarks: random higher-order PUBO instances and MAX-3-SAT (via the standard PUBO mapping) and compare against standart algorithm GNN solvers, including QUBO-based that solve problem after the quadratization procedure. Across settings, PB-HGNN attains competitive or superior solution quality while remaining simple and scalable.

**Contributions.**

- Direct higher-order modeling for UL. We propose PB-HGNN, the first unsupervised neural framework that solves PUBO *without* quadratization, capturing $k$-way structure directly.

- Memory-driven architecture. We utlize a compact epoch-level memory that carries information across iterations, stabilizing instance-specific training with minimal overhead.

- Empirical validation. On random PUBO and MAX-3-SAT, PB-HGNN matches or surpasses quadratization-based GNN baselines.

## 2 COMBINATORIAL OPTIMIZATION WITH UNSUPERVISED LEARNING

**Unsupervised Learning (UL).** Neural approaches to CO include supervised prediction Prates et al. (2019); Gasse et al. (2019) and reinforcement learning (RL) Khalil et al. (2017); Kool et al. (2019); supervised methods require labeled near-optimal solutions and often struggle to generalize, whereas RL avoids labels but is often sample-inefficient on large state spaces. A complementary route is unsupervised, instance-specific learning: optimize a differentiable model *through* which the decision variables are produced, and update model parameters to improve the instance's objective, rather than updating the variables directly. This reparameterization often yields smoother search trajectories and fewer stalls in poor local minima Wang et al. (2022), and, by training on the instance itself, encodes useful inductive bias without requiring labeled solutions. In effect, UL turns the solver into a trainable generator of candidates that can be optimized end-to-end with standard gradient methods.

**GNN-based UL** Many CO problems are naturally defined on graphs, where solutions select subsets of nodes/edges; this makes graph neural networks appealing as the learnable maps that produce instance-specific decisions Cappart et al. (2021). Unsupervised GNN solvers include recurrent architectures for constrained satisfaction problems such as RUN-CSP Tönshoff J & M (2021); instance-specific training for SAT Amizadeh et al. (2018); and Erdős Goes Neural (EGN) Karalias & Loukas (2020), which learns distributions over solutions inspired by the probabilistic method, later extended via entry-wise concave relaxations Wang et al. (2022) and meta-learning variants. Min et al. (2023) train GNN with a surrogate unsupervised loss to produce an edge 'heat map', and a local search then decodes a TSP tour from it. Heydaribeni et al. (2024) first proposed using hypergraph neural networks for unsupervised combinatorial optimization. Their method, HypOp, models each constraint as a hyperedge in a constraint hypergraph and trains a HyperGNN as a learnable transformation function, with a simulated annealing step for mapping continuous relaxations back to discrete solutions. These methods differ in how they parametrize decisions, which unsupervised signals they optimize (constraint-violation penalties, differentiable surrogates), and how they control discretization (e.g., temperature schedules or annealing), but all share the goal of learning instance-specific heuristics directly from the objective without labeled solutions.

**QUBO** A prominent UL line reduces the target CO to QUBO, which is the problem to minimize a pseudo-Boolean polynomial 1 $f(x)$ of degree two Hammer (1968):

$$\min_{x \in \{0,1\}^n} x^T Q x, \tag{2}$$

PI-GNN Schuetz et al. (2022a) replaces binary variables by probabilities $p_\theta \in [0,1]^n$ output by a GNN and uses the relaxed QUBO as a differentiable loss, enabling general solvers whenever a QUBO formulation exists; follow-ups adapt this paradigm to specific tasks (e.g., graph coloring Schuetz et al. (2022b); Wang et al. (2023)). To direct relaxation toward discrete solutions while keeping a smooth landscape, Ichikawa (2024) proposes a principled annealing schedule by loss regularization, that first facilitates exploration and then promotes automatic rounding. Pugacheva et al. (2024) proposed the novel GNN architecture design that feeds model outputs back into the next forward pass: Apply previous node states as dynamic node features and concatenate them with static features across epochs, improving stability and convergence of the UL procedure. Our work follows this UL paradigm but extends it to *hypergraphs* to natively capture higher-order terms in the objective function.

## 3 PRELIMINARIES: HYPERGRAPH NEURAL NETWORKS

A hypergraph $\mathcal{H} = (V, E)$ generalizes a graph by allowing each hyperedge $e \in E$ to connect an arbitrary subset of nodes $e \subseteq V$. This structure naturally captures higher-order relations that are not representable with simple pairwise edges and is widely used across machine learning and network science Kim et al. (2024); Estrada (2005). Let $|V| = n$ and $|E| = m$. We write the binary incidence matrix $A \in \{0,1\}^{n \times m}$ with $A_{i,e} = 1$ iff $i \in e$, and define node and hyperedge degrees by $d_v(i) = \sum_e A_{i,e}$ and $d_e(e) = \sum_i A_{i,e}$, respectively. When hyperedges carry attributes (e.g., weights, types), we denote them by $a_e$.

**Message passing on hypergraphs.** Hypergraph Neural Networks (HGNNs) extend GNN message passing by alternating *node→edge* and *edge→node* aggregations Feng et al. (2019). At layer $\ell$, with node features $X^{(\ell)} \in \mathbb{R}^{n \times d_\ell}$, a general two-step update can be written as

$$(\text{node} \to \text{edge}) \quad U_e^{(\ell)} = \text{AGG}_e \big\{ \phi_n^{(\ell)}(X_i^{(\ell)}) : i \in e \big\},$$

$$(\text{edge} \to \text{node}) \quad X_i^{(\ell+1)} = \sigma \Big( \psi_n^{(\ell)} \big( \bigoplus_{e \ni i} \phi_e^{(\ell)}(U_e^{(\ell)}, a_e) \big) \Big),$$

where $\phi_n, \phi_e, \psi_n$ are learnable maps, $\text{AGG}_e$ and $\oplus$ are permutation-invariant set aggregators (e.g., mean/sum), and $\sigma$ is an activation. This template covers a broad family of HGNNs: different normalizations, stabilizers (e.g., BatchNorm), and choices of aggregators/MLPs lead to different concrete architectures Feng et al. (2019); Kim et al. (2024).

**A common instantiation (mean/sum).** A widely used choice (which we adopt further) applies linear maps before each hop and uses mean on the node→edge step and sum on the edge→node step:

$$U_e^{(\ell)} = \frac{1}{|e|} \sum_{i \in e} W_1^{(\ell)} X_i^{(\ell)}, \qquad X_i^{(\ell+1)} = \sigma\Big(\text{BN}\big(\sum_{e \ni i} W_2^{(\ell)} U_e^{(\ell)}\big)\Big),$$

with trainable $W_1^{(\ell)} \in \mathbb{R}^{d_\ell \times d_h}$, $W_2^{(\ell)} \in \mathbb{R}^{d_h \times d_{\ell+1}}$, and an optional BatchNorm ("BN") for stability. Hyperedge attributes can modulate messages, e.g., by multiplying each $U_e^{(\ell)}$ with a scalar weight $w_e$ or with learned embeddings Dong et al. (2020); Chien et al. (2022). Stacking two such hops with a residual (skip) connection yields a residual hypergraph block:

$$\widehat{X}^{(\ell+1)} = \text{BN}\big(\sum_{e \ni i} W_2^{(\ell)} U_e^{(\ell)}\big), \quad X^{(\ell+1)} = \sigma\big(\widehat{X}^{(\ell+1)} + \text{Skip}(X^{(\ell)})\big).$$

Alternative normalizations based on degree matrices (e.g., variants derived from $D_v$ and $D_e$) are also common and can be plugged into either hop to control scale and improve conditioning Zhou et al. (2006). The generic template above is agnostic to these choices and encompasses classical HGNNs and related approximations.

# 4 PB-HGNN: HYPERGRAPH NEURAL NETWORKS FOR PSEUDO-BOOLEAN OPTIMIZATION

In this section, we describe a detailed explanation of the proposed framework.

**Hypergraph Construction.** Given a pseudo-Boolean polynomial 1 of degree $d \geq 3$, we construct a hypergraph $\mathcal{H} = (\mathcal{V}, \mathcal{E})$ where $\mathcal{V} = \{v_1, v_2, \ldots, v_n\}$ represents the set of variables $\{x_1, x_2, \ldots, x_n\}$ and $\mathcal{E} = \{e_S : S \subseteq \{1, 2, \ldots, n\}, c_S \neq 0\}$ represents hyperedges, weighted by $w_S = c_S$, corresponding to polynomial terms $c_S \prod_{i \in S} x_i$. An illustrative example on a small pseudo-Boolean polynomial is shown in Figure 1. This representation preserves the exact structure of higher-order dependencies.

**Continuous Relaxation.** In the literature on unsupervised learning for QUBO (e.g., PI-GNN and related approaches), it is common to replace the binary decision variables $x_i \in \{0, 1\}$ by continuous probabilities $p_i \in [0, 1]$. A neural network is then trained to output these probabilities, which represent the likelihood that variable $i$ is assigned the value 1. The resulting continuous objective is

$$\mathcal{L}(p) = \sum_{S \subseteq [n]} c_S \prod_{i \in S} p_i, \qquad p \in [0, 1]^n. \tag{3}$$

While this relaxation is usually motivated from a machine learning perspective, it coincides exactly with the *multilinear extension* Calinescu et al. (2007) of the underlying pseudo-Boolean function. This connection provides both an algebraic and a probabilistic interpretation of $\mathcal{L}(p)$.

**Lemma 1** (Multilinear extension of pseudo-Boolean polynomials). *Let $f : \{0, 1\}^n \to \mathbb{R}$ be a pseudo-Boolean polynomial 1. Then its multilinear extension coincides with $\mathcal{L}(p)$ in equation 3.*

The multilinear extension admits a natural probabilistic view: if $X = (X_1, \ldots, X_n)$ with $X_i \sim \text{Bernoulli}(p_i)$ independently, then

$$\mathcal{L}(p) = \mathbb{E}[f(X)].$$

Hence the relaxed objective evaluates the expected value of the original discrete function under independent randomized rounding with marginals $p$. This interpretation is crucial in practice: optimizing $\mathcal{L}(p)$ corresponds to searching for probability distributions over feasible solutions that yield high-quality discrete outcomes upon sampling or rounding.

The continuous variables are parametrized by a hypergraph neural network: $p = \sigma(\mathcal{F}_\theta(\mathcal{H}, X^{(0)}))$ where $\sigma$ is the sigmoid function, $\mathcal{F}_\theta$ is the HGNN with parameters $\theta$, and $X^{(0)} \in \mathbb{R}^{n \times d_0}$ are node embeddings.

Figure 2: The PB-HGNN framework high-level design. (1) *PUBO Instance:* the input is a pseudo-Boolean polynomial and its probabilistic extension used for training. (2) *Hypergraph Construction:* each nonzero monomial becomes a hyperedge on its variables with weight equal to the coefficient. (3) *PB-HGNN:* a hypergraph neural network predicts probabilities by minimizing probabilistic extension. (4) *Solution:* After obtaining probabilities on the final iteration (epoch) $T$, a discrete assignment is obtained by thresholding.

**Unsupervised Learning.** The full PB-HGNN training framework is illustrated in Figure 3. Starting from the pseudo-Boolean polynomial, we construct the hypergraph $\mathcal{H} = (\mathcal{V}, \mathcal{E})$ as described above. Each node $v_i \in \mathcal{V}$ is initialized with an embedding $X_i^{(0)} \in \mathbb{R}^{d_0}$, drawn from a normalized random distribution or from problem-specific features if available. These embeddings are propagated through a stack of hypergraph convolutional and residual layers,

$$X^{(\ell+1)} = \phi_\ell\big(X^{(\ell)}, \mathcal{H}; \theta_\ell\big),$$

where $\phi_\ell$ denotes the $\ell$-th HGNN layer parameterized by $\theta_\ell$. After $L$ layers, we obtain node-level hidden states $X^{(L)}$. A final projection layer followed by a sigmoid activation produces probabilities

$$p \;=\; \sigma\big(X^{(L)} W_{\text{out}} + \mathbf{1}\, b_{\text{out}}\big) \;\in [0,1]^n, \qquad X^{(L)} \in \mathbb{R}^{n \times d_L}, \;\; W_{\text{out}} \in \mathbb{R}^{d_L \times 1}, \;\; b_{\text{out}} \in \mathbb{R}.$$

These probabilities are interpreted as soft assignments of binary variables. The loss $\mathcal{L}(p)$ is computed directly from $p$ and the polynomial coefficients, as detailed in the previous subsection.

Training proceeds in an unsupervised manner: given $p$, we evaluate the loss $\mathcal{L}(p)$, perform backpropagation through all layers of the network, and update the HGNN parameters $\theta = \{\theta_\ell, W_{\text{out}}, b_{\text{out}}\}$ using stochastic gradient descent (AdamW in our implementation). This iterative process continues until convergence or early stopping, yielding a trained PB-HGNN that maps a problem instance to a set of variable probabilities. After that, probabilities $p$ are projected to a discrete solution by applying an indicator function,

$$x_i = \mathbf{1}\{\, p_i > \tau \,\}, \qquad \tau \in (0,1),$$

with $\tau = 0.5$ as the standard threshold. This simple rounding yields a binary assignment that can optionally be refined by local search.

## 4.1 PB-HGNN ARCHITECTURE

**Memory-augmented hypergraph convolution.** The performance of unsupervised neural solvers for combinatorial optimization depends critically on the underlying GNN architecture. In PB-HGNN we employ a memory-augmented hypergraph convolution block as the core unit. Alongside the static node embeddings $X^{(0)} \in \mathbb{R}^{n \times d_0}$, each node $i$ carries a compact hidden memory $m_i^t \in \mathbb{R}^{d_m}$ ($d_m \ll d$), and we collect these row-wise into $M^t \in \mathbb{R}^{n \times d_m}$. At epoch $t$, the block takes $\big[\, X^{(0)} \,\|\, M^t \,\big]$ as input, performs hypergraph message passing with a residual (skip) connection, produces probabilities $p^t$, and updates the memory by projecting the current hidden states:

$$m_i^{t+1} \;=\; \tanh\big(W_m\, h_i^t\big), \qquad W_m \in \mathbb{R}^{d \times d_m}.$$

where $h_i^t \in \mathbb{R}^d$ denotes the block's output representation of node $i$ at epoch $t$, and $W_m \in \mathbb{R}^{d \times d_m}$. We detach $M^{t+1}$ from the computation graph before the next epoch, preventing backpropagation across epochs and keeping the per-epoch training cost stable. Thus the model refines its predictions over training iterations without recomputing from scratch.

Conceptually, our design is a lightweight variant of the iterative refinement strategy Pugacheva et al. (2024). In QI-GNN, refinement is performed across epochs by feeding the previous epoch's predicted probabilities back into the model as additional node features. PB-HGNN follows the same high-level idea to carry information across epochs, but does so via a compact hidden memory: instead of using raw predictions, we propagate a learned state $M^t$ obtained from the current hidden representations and detached before the next epoch. This keeps the signal low-dimensional, reduces noise from raw outputs, and yields a simple, stable refinement mechanism embedded in the architecture.

---

**Algorithm 1** PB-HGNN: one-epoch forward/backward with epoch-level memory

---

1: **Input:** hypergraph $\mathcal{H} = (V, E)$; node features $X^{(0)} = \{x_i^{(0)}\}$; weights $\{w_e\}$; incidence $V(e), E(i)$; order emb. $\phi(\cdot)$; memory $M^t = \{m_i^t\}$
2: **Output:** probabilities $P^t = \{p_i^t\}$; next memory $M^{t+1} = \{m_i^{t+1}\}$
3: $h_i^{0,t} \leftarrow [\, x_i^{(0)} \| m_i^t\,]$; $\quad z_i^t \leftarrow \text{ReLU}(W_{\text{in}} h_i^{0,t})$ $\hfill \triangleright$ input fusion & projection
4: $u_e^t \leftarrow \frac{1}{|e|} \sum_{i \in V(e)} W_1 z_i^t$ $\hfill \triangleright$ node→edge: mean aggregation
5: $\tilde{a}_i^t \leftarrow \sum_{e \in E(i)} \left[ W_2(u_e^t \odot \phi(|e|)) \right] \cdot w_e$ $\hfill \triangleright$ edge→node: sum & modulation
6: $a_i^t \leftarrow \text{ReLU}(\text{BN}_1(\tilde{a}_i^t))$ $\hfill \triangleright$ norm & activation
7: $u_e'^t \leftarrow \frac{1}{|e|} \sum_{i \in V(e)} W_1' a_i^t$ $\hfill \triangleright$ second pass: node→edge
8: $\tilde{b}_i^t \leftarrow \sum_{e \in E(i)} \left[ W_2'(u_e'^t \odot \phi(|e|)) \right] \cdot w_e$ $\hfill \triangleright$ second pass: edge→node
9: $b_i^t \leftarrow \text{BN}_2(\tilde{b}_i^t)$; $\quad h_i^t \leftarrow \text{ReLU}(b_i^t + \text{Skip}(z_i^t))$ $\hfill \triangleright$ residual (skip)
10: $p_i^t \leftarrow \sigma(w_{\text{out}}^\top h_i^t + b_{\text{out}})$ $\hfill \triangleright$ output head (sigmoid)
11: $\mathcal{L} \leftarrow \sum_S c_S \prod_{j \in S} p_j^t$ $\hfill \triangleright$ probabilistic relaxation loss
12: **update** $\Theta$ with AdamW using $\nabla_\Theta \mathcal{L}$ $\hfill \triangleright$ backward
13: $m_i^{t+1} \leftarrow \tanh(W_m h_i^t)$; **detach** all $m_i^{t+1}$ $\hfill \triangleright$ memory update for next epoch
14: **return** $P^t, M^{t+1}$

---

We use a single residual hypergraph–convolution block whose state is carried *across epochs* via a compact per-node memory. For selected epoch, we show Algorithm 1 with details of the forward/backword procedure. Alongside the static embeddings $X^{(0)} \in \mathbb{R}^{n \times d_0}$, each node $i$ keeps a memory vector $m_i^t \in \mathbb{R}^{d_m}$ ($d_m \ll d$); we stack them row-wise as $M^t$ for input fusion. In each epoch $t$, the block (i) concatenates features with memory and projects to the working width (line 3); (ii) performs a two-hop message pass with *mean* node→edge aggregation (line 4) and *sum* edge→node aggregation modulated by the hyperedge order embedding $\phi(|e|)$ and polynomial weight $w_e$ (lines 5–6); (iii) repeats the two hops with independent parameters and applies BatchNorm and a residual (skip) connection (lines 7–9). The output head produces per-node probabilities via a sigmoid (line 10), and the loss is the multilinear extension of the original polynomial evaluated at these probabilities (line 11). We update parameters with AdamW (line 12), then update the per-node memories by a tanh projection of the current hidden states and *detach* them before the next epoch to avoid backpropagation through epochs (line 13). This realizes an iterative refinement mechanism across epochs through a lightweight, learned hidden state rather than by feeding back raw predictions.

**Training details.** By default, we use a single residual hypergraph–convolution block with Batch-Norm and ReLU, a linear output head with a sigmoid, and a compact per-node memory of size $d_m = \max\{1, \lfloor d/8 \rfloor\}$, where $d$ is the working hidden width. Node features $X^{(0)} \in \mathbb{R}^{n \times d_0}$ are *static*: unless problem-specific features are provided, we initialize them with $\ell_2$-normalized Gaussian vectors. Hyperedges are stored in a sparse incidence format and carry (i) the polynomial weight $w_e = c_S$ and (ii) an order embedding $\phi(|e|)$; in our implementation we cap the order vocabulary at $|e| \leq 5$ for efficiency (higher orders are clamped). We optimize parameters $\theta$ with AdamW, apply gradient clipping (max-norm 0.5), and use a REDUCELRONPLATEAU scheduler; early stopping is triggered when the loss stabilizes. Mixed precision is enabled when available. The memory state is propagated *across epochs* and *detached* before the next epoch to avoid backpropagation through time. At inference, we decode by thresholding $x_i = \mathbf{1}\{p_i > \tau\}$ with default $\tau = 0.5$. For efficiency, we pre-compile loss terms by degree (vectorized for $d = 1$, batched for $d = 2$, tensorized

for $d \geq 3$). While the architecture naturally supports deeper stacks and alternative memory updates, our experiments use this minimal configuration for speed and stability.

## 5 NUMERICAL EXPERIMENTS

### 5.1 RANDOM POLYNOMIALS

We consider synthetic pseudo-Boolean objectives of fixed degree $k \geq 3$ with $n$ binary variables. Following prior work Chermoshentsev et al. (2021), we generate three standard classes of random instances: (i) *Class I (dense $\pm 1$):* each $k$-tuple coefficient $J_{i_1 \ldots i_k}$ is sampled i.i.d. from $\{\pm 1\}$ with equal probability - it corresponds to the extended version of Sherrington–Kirkpatrick model. Chermoshentsev et al. (2021). (ii) *Class II (dense uniform):* $J_{i_1 \ldots i_k} \sim \text{Unif}[-1, 1]$ i.i.d.; (iii) *Class III (sparse):* start from Class II and independently drop each coefficient with probability 0.9 (set it to zero), producing a sparse higher-order interaction tensor.

Let $x \in \{0, 1\}^n$. A degree-$k$ random polynomial reads

$$p_k(x) = \sum_{1 \leq i_1 < \cdots < i_k \leq n} J_{i_1 \ldots i_k} \, x_{i_1} \cdots x_{i_k}, \tag{4}$$

with coefficients $J_{i_1 \ldots i_k}$ drawn according to the chosen class. When needed, we also include lower-order terms to allow affine shifts:

$$p(x) = C_{\emptyset} + \sum_i C_i x_i + \sum_{i<j} C_{ij} x_i x_j + \sum_{i_1 < \cdots < i_k} J_{i_1 \ldots i_k} x_{i_1} \cdots x_{i_k}. \tag{5}$$

The optimization task is $\min_{x \in \{0,1\}^n} p(x)$ (or equivalently, maximization with flipped signs). This is a native PUBO instance with hyperedges $e_{\{i_1, \ldots, i_k\}}$ weighted by $J_{i_1 \ldots i_k}$. **?**

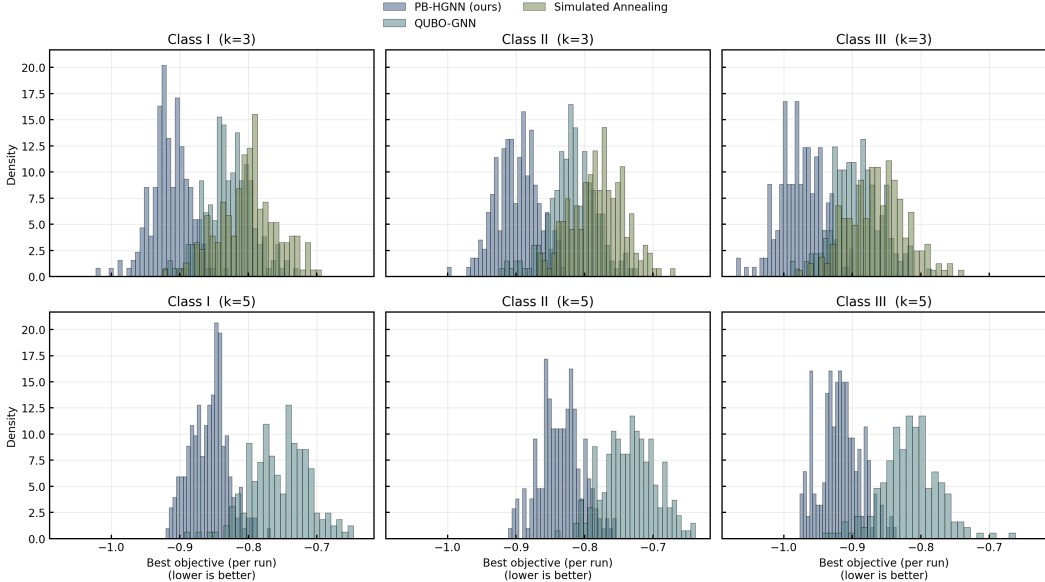

Figure 3: Outcome distributions at $k = \{3, 5\}$ (PB-HGNN vs. SA). Columns correspond to the three random PUBO classes, Each panel aggregates $R=100$ independent runs (different random initializations) and shows the density of the *normalized* best objective reached within the same compute budget. We omit QUBO GNNs for k = 5 since it requires very large number of variables (tens thousands) in the QUBO reformulation

**Experimental Setup and Results.** We study random PUBO instances at fixed size $n=100$ and degrees $k \in \{3, 5\}$ for the three classes, as defined above(. For each $(k, \text{class})$ we generate 10

independent instances and, unless noted, report aggregates across these instances. PB-HGNN is trained for 10,000 epochs with AdamW. We evaluate the discrete objective $p(x)$ each epoch, and keep the best-so-far value. As baselines, we use (i) a QUBO-GNN trained on a quadratized version of the same instances with the same epoch budget and optimizer, and (ii) Simulated Annealing (SA) on the PUBO form.

We visualize results with three figure sets. We show *Outcome distributions* 3 ($2\times3$ grid): for each panel we aggregate the final best-of-run objectives over $R=100$ runs (multiple random initializations) and display density histograms for PB-HGNN, QUBO-based GNN (that solves the QUBO problem after quadratization) : we select Pugacheva et al. (2024) as it acheives the best performance among QUBO GNNs, and Simulated Annealing, revealing both spread and tail behavior. To make values comparable across instances with different scales and sparsities, we report $\widetilde{p}(x) = p(x)/\|c\|_1$, where $\|c\|_1 = \sum_S |c_S|$ is the $\ell_1$ norm of the PUBO coefficients for that instance; lower is better. This scale-free normalization removes instance-dependent magnitude effects (e.g., number of nonzeros, coefficient ranges), so the histograms reflect solver quality rather than raw objective scale. Across all classes, SA is competitive and lies close to PB-HGNN under this normalization, indicating that SA is not drastically worse at $k=5$. Across classes and both degrees, PB-HGNN consistently reaches lower objectives faster than the QUBO baselines and exhibits tighter outcome distributions; the overhead plots highlight the rapidly growing variable/term counts for higher $k$, underscoring the advantage of operating natively on the PUBO hypergraph.

## 5.2 MAX-3-SAT

**Problem description.** Given Boolean variables $y_1, \ldots, y_n \in \{0, 1\}$ and a 3-CNF formula with $m$ clauses $C_r = (\ell_{r1} \vee \ell_{r2} \vee \ell_{r3})$, each literal $\ell_{rk}$ being either a variable $y_{i_{rk}}$ or its negation $\neg y_{i_{rk}}$, the goal in MAX-3-SAT is to assign truth values to the variables so as to maximize the number of satisfied clauses. A clause is satisfied if at least one of its three literals is true, and violated if all three are false. The canonical objective is therefore

$$\max_{y \in \{0,1\}^n} \sum_{r=1}^m \mathbf{1}\{C_r(y) = \text{True}\} \; = \; m - \min_{y \in \{0,1\}^n} \sum_{r=1}^m \mathbf{1}\{C_r(y) = \text{False}\}.$$

We consider the equivalent minimization of the number of violated clauses (right-hand side), which is convenient for our PUBO mapping. MAX-3-SAT is NP-hard and widely used as a benchmark for discrete optimization; instances are often specified by their clause density $\alpha = m/n$ and may be *weighted* (each clause has a weight) or *unweighted* (all weights $= 1$). Performance is typically reported as the fraction of satisfied clauses or the number of violations at the returned assignment.

**PUBO formulation.** Let $x_i \in \{0, 1\}$ encode $y_i = \text{True} \iff x_i = 1$. For a literal $\ell$ on $x_i$, define its falsity factor

$$\overline{\ell}(x) = \begin{cases} 1 - x_i, & \ell = x_i, \\ x_i, & \ell = \neg x_i. \end{cases}$$

A 3-clause $C_r = (\ell_{r1} \vee \ell_{r2} \vee \ell_{r3})$ is violated iff all three literals are false, hence

$$v_r(x) = \prod_{k=1}^3 \overline{\ell_{rk}}(x).$$

Thus MAX-3-SAT becomes the unconstrained pseudo-Boolean optimization

$$\min_{x \in \{0,1\}^n} p(x) = \sum_{r=1}^m v_r(x) \; = \; C_\emptyset + \sum_i C_i x_i + \sum_{i<j} -C_{ij} x_i x_j + \sum_{i<j<k} C_{ijk} x_i x_j x_k,$$

where the coefficients $\{C_\emptyset, C_i, C_{ij}, C_{ijk}\}$ follow from expanding the products and collecting like terms. (Full coefficient formulas and derivation are provided in Appendix B.)

**Experimental Setup and Results.** We evaluate PB-HGNN on the MAX-3-SAT problem following the experimental design of Yau et al. citeyau2024are. We run Two settings are considered: (i) random unweighted instances with $n = 100$ variables and clause density $\alpha = m/n \in$

Table 1: Average number of unsatisfied clauses for Max-3-SAT on random instances with $N = 100$ variables and $\alpha \in \{4.00, 4.15, 4.30\}$. Standard deviation is shown after $\pm$. In parentheses: average runtime per instance (seconds).

| Method | $\alpha = 4.00$ | $\alpha = 4.15$ | $\alpha = 4.30$ |
|---|---|---|---|
| WalkSAT | 0.94±0.92 | 1.46±1.11 | 1.97±1.28 |
| Survey Propagation | 3.32±0.81 | 3.87±0.79 | 3.94±0.93 |
| ErdősGNN | 5.46±1.91 | 6.14±2.01 | 6.79±2.03 |
| OptGNN | 4.46±1.68 | 5.15±1.76 | 5.84±2.18 |
| PB-HGNN (ours) | 3.11±0.98 | 4.01±1.10 | 4.29±1.45 |

$\{4.00, 4.15, 4.30\}$, and (ii) benchmark instances from the UUF SATLIB dataset. In both cases, the objective is to minimize the number of violated clauses.

For the random instance setting, each formula is generated on the fly with uniformly random clauses, and results are averaged over multiple test instances. We directly replicate the baselines reported in Yau et al. (2024): *WalkSAT* (classic stochastic local search solver), *Survey Propagation* Braunstein et al. (2005), *ErdősGNN* Karalias & Loukas (2020), and unsupervised OptGNN Yau et al. (2024). We then add a row for our *PB-HGNN*. Performance is measured as the average number of unsatisfied clauses (lower is better). The results show that our method outperforms both neural solvers, while being inferiour to the best classical problem-specific method.

For the second setting, we evaluate on the UUF benchmark SAT instances from SATLIB, commonly used to test incomplete Max-SAT solvers. We select 100 unsatisfiable instances from uuf125.538.100 dataset, with problems of 100 variables and 538 clauses, and report the average fraction of satisfied clauses across instances. We compare with local search sat-specific heuristics WalkSAT and Heuristic Backbone Sampling (HBS), and neural baseline *ErdősGNN* Karalias & Loukas (2020).

Table 2: Performance on UUF benchmark instances from SATLIB. We report average clause satisfaction (higher is better). PB-HGNN results will be added in the last row.

| Method | average, satisfied clauses |
|---|---|
| WalkSAT | 536.42 |
| HBS | 536.11 |
| ErdősGNN | 528.75 |
| PB-HGNN (ours) | 534.30 |

## 6 CONCLUSIONS

We introduced PB-HGNN, an unsupervised hypergraph–neural framework that optimizes pseudo-Boolean polynomials directly in their native higher-order form. By mapping each nonzero monomial to a hyperedge with the corresponding coefficient, and training a memory-augmented HGNN to minimize the probabilistic relaxation, our method manages to capture multi-way variable interactions. Across representative benchmark families: random higher-order PUBO, including the Sherrington–Kirkpatrick spin model, and MAX-3-SAT, PB-HGNN delivers competitive or superior solution quality to GNN baselines built on reduced quadratic formulations. heuristics, while scaling to thousands of variables. These results demonstrate that hypergraph message passing paired with instance-specific unsupervised training is a novel and effective recipe for higher-order combinatorial optimization.

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

## A  PROOFS

**Lemma 1.** (Multilinear extension of pseudo-boolean polynomial). *Let $f : \{0,1\}^n \to \mathbb{R}$ be a pseudo-Boolean polynomial*

$$f(x) = \sum_{S \subseteq [n]} c_S \prod_{i \in S} x_i, \qquad x \in \{0,1\}^n. \tag{6}$$

*Then its multilinear extension coincides with $\mathcal{L}(p)$ in equation 3*, i.e.,

$$F(p) = \sum_{S \subseteq [n]} c_S \prod_{i \in S} p_i, \qquad p \in [0,1]^n.$$

*Proof.* By definition, the multilinear extension of a set function $f$ is

$$F(p) = \sum_{S \subseteq [n]} f(S) \prod_{i \in S} p_i \prod_{i \notin S} (1 - p_i).$$

Evaluating $f(S)$ from its polynomial form gives

$$f(S) = f(\mathbf{1}_S) = \sum_{T \subseteq [n]} c_T \prod_{j \in T} \mathbf{1}_S(j) = \sum_{T \subseteq S} c_T.$$

Substituting into $F(p)$ yields

$$F(p) = \sum_{S \subseteq [n]} \Big( \sum_{T \subseteq S} c_T \Big) \prod_{i \in S} p_i \prod_{i \notin S} (1 - p_i).$$

Swapping the order of summation gives

$$F(p) = \sum_{T \subseteq [n]} c_T \sum_{S \supseteq T} \prod_{i \in S} p_i \prod_{i \notin S} (1 - p_i).$$

For each fixed $T$, factor out $\prod_{i \in T} p_i$ and write $U = S \setminus T$:

$$\sum_{S \supseteq T} \prod_{i \in S} p_i \prod_{i \notin S} (1 - p_i) = \Big( \prod_{i \in T} p_i \Big) \sum_{U \subseteq [n] \setminus T} \prod_{i \in U} p_i \prod_{i \notin U} (1 - p_i).$$

The inner sum is the expansion of $\prod_{i \in [n] \setminus T} (p_i + (1 - p_i)) = 1$. Hence

$$F(p) = \sum_{T \subseteq [n]} c_T \prod_{i \in T} p_i,$$

which equals $\mathcal{L}(p)$ in equation 3. □

### A.1 HYPERPARAMETER SETTINGS

Default hyperparameters optimized through extensive experiments: embedding dimension $d = 16$, hidden dimension $d_h = 8$, memory dimension $d_m = 2$, number of layers $L = 2$, learning rate $\alpha = 10^{-3}$ with ReduceLROnPlateau scheduling, dropout rate $\rho = 0.05$, weight decay $\lambda_2 = 10^{-6}$.

### A.2 EFFICIENT LOSS COMPUTATION

Naively evaluating equation 3 costs $O(n^d)$. We pre-compile terms by degree:

$$\mathcal{L}(p) = \sum_{d=0}^{D} \mathcal{C}[d](p), \tag{7}$$

where: - $\mathcal{C}[1](p) = \sum_i c_i p_i$ (vectorized), - $\mathcal{C}[2](p) = \sum_{i,j} c_{ij} p_i p_j$ (batched matrix ops), - $\mathcal{C}[d](p) = \sum_{S:|S|=d} c_S \prod_{i \in S} p_i$ for $d \geq 3$ (tensorized).

The hypergraph is stored in sparse index format

$$\mathcal{E}_{\text{sparse}} = \{(i, e) : i \in e\},$$

reducing complexity from $O(nm)$ to $O(|\mathcal{E}_{\text{sparse}}|)$.

## B DERIVATION DETAILS FOR MAX 3 SAT PUBO

For a literal on variable $x_i$, let the negation bit $\alpha \in \{0, 1\}$ indicate its sign ($\alpha = 0$ for $x_i$, $\alpha = 1$ for $\neg x_i$). Define $s := 1 - 2\alpha \in \{+1, -1\}$. Then the falsity factor can be written as

$$\bar{\ell}(x) = (1 - \alpha) - s\, x_i.$$

For a clause $C_r = (\ell_{r1} \vee \ell_{r2} \vee \ell_{r3})$ on indices $I_r = \{i_{r1}, i_{r2}, i_{r3}\}$ with signs $\{\alpha_{r1}, \alpha_{r2}, \alpha_{r3}\}$ and $s_{rk} = 1 - 2\alpha_{rk}$, the violation indicator expands as

$$v_r(x) = \prod_{k=1}^{3} \left((1 - \alpha_{rk}) - s_{rk} x_{i_{rk}}\right) = c_\emptyset^{(r)} + \sum_{p \in I_r} c_p^{(r)} x_p + \sum_{\{p,q\} \subset I_r} c_{pq}^{(r)} x_p x_q + c_{pqr}^{(r)} x_p x_q x_r,$$

with coefficients

$$c_\emptyset^{(r)} = \prod_{k=1}^{3} (1 - \alpha_{rk}),$$

$$c_{i_{rk}}^{(r)} = -s_{rk} \prod_{\ell \neq k} (1 - \alpha_{r\ell}),$$

$$c_{i_{rk} i_{r\ell}}^{(r)} = s_{rk}\, s_{r\ell} \prod_{j \neq k,\ell} (1 - \alpha_{rj}),$$

$$c_{i_{r1} i_{r2} i_{r3}}^{(r)} = -s_{r1} s_{r2} s_{r3}.$$

Summing over clauses yields

$$p(x) = \sum_{r=1}^{m} v_r(x) = C_\emptyset + \sum_i C_i x_i + \sum_{i<j} C_{ij} x_i x_j + \sum_{i<j<k} C_{ijk} x_i x_j x_k,$$

where $C_\emptyset = \sum_r c_\emptyset^{(r)}$, $C_i = \sum_r c_i^{(r)}$, $C_{ij} = \sum_r c_{ij}^{(r)}$, and $C_{ijk} = \sum_r c_{ijk}^{(r)}$. Equivalently, one may enumerate all $2^3$ sign patterns of a clause and record the resulting monomials; both approaches are algebraically identical.

