# OpenReview forum: "Optimizing Pseudo-Boolean Polynomials using Hypergraph Neural Networks"
_ICLR.cc/2026/Conference — ICLR 2026 Conference Withdrawn Submission_

### Official Review · Reviewer_sRHc · 2025-10-30

**Soundness:** 2
**Presentation:** 3
**Contribution:** 1
**Rating:** 4
**Confidence:** 4

**Summary:**

This work falls under GNNs for optimization (a factor-graph–style message passing where aggregation and nonlinear maps are learned by minimizing a loss, rather than fixed BP updates from a generative model). The proposed method starts from a PUBO, use its multilinear extension as a continuous relaxation and optimize that loss. It initializes a continuous state then runs the hypergraph message passing with residual connections and a compact cross-epoch memory that is updated each epoch. Training is unsupervised and instance-specific for the parameters of the GNN using SGD.

Using GNNs for CO is not new; the novel contribution here is to targets higher-order interactions via hypergraphs for PUBO problems.

**Strengths:**

The main strength is that the framework extends GNN-for-optimization to higher-order structure using hypergraphs beyond-pairwise (although HypOp by Heydaribeni et al. Nature Machine Intelligence already developed the concept).
The paper presents a sound benchmark (higher order SK and Max-3-SAT mapped to Ising). High-level concepts are introduced clearly for a general reader. The method is well explained with clear pseudocode.

**Weaknesses:**

The benchmarking is too limited to be able to judge whether the approach is truly better than SOTA heuristics.
The results section omits runtime, it seems. Without time, it is hard to judge practical effectiveness, e.g., time to a target gap or time-to-target quality.

The paper includes several typos suggest the benchmark and write-up need polishing.

**Questions:**

1) Table 1 states “In parentheses: average runtime per instance (seconds),” but no times are shown. Without runtimes, it is hard to assess effectiveness. Please provide wall clock time for comparison, together with hardware used.

2) Figure 3 reports results under “same compute budget,” but seconds and hardware are not provided.
How long is SA run? What is the SA runtime vs the hypergraph method? Please report time and schedule/iteration budget.

3) For Max-SAT, how do results compare to SOTA SAT/Max-SAT solvers in time-to-target?

4) How does the method scale with higher-order terms and mixed orders?
In very inhomogeneous cases, how much does clamping higher-order terms affect performance? Is the method limited to regular instances?

5) How does the memory depth of cross-epoch memory affect performance? After how many epochs do gains vanish?

6) Typos around line 348 and line 430.

---

### Official Review · Reviewer_SB5J · 2025-10-30

**Soundness:** 2
**Presentation:** 1
**Contribution:** 2
**Rating:** 2
**Confidence:** 4

**Summary:**

The paper claims to be the first unsupervised neural framework that solves unconstrained polynomial optimization without “quadratization”.  The primary idea is simple, replace the monomials of the polynomial with edges of a hypergraph and train a hypergraph neural network on the resulting optimization problem completely unsupervised.  The paper compares their approach PB-HGNN to QUBO-GNN, simulated annealing for random polynomial optimization instances.  They also compare against various neural baselines and classical benchmarks like WalkSAT on SATLIB.

**Strengths:**

Unconstrained polynomial optimization is a very broad class of problems.  The approach to solving the problem is very natural.  I quite like the criticism of quadratization.

**Weaknesses:**

These are not all weaknesses, but I figure I’d put the nuanced review in this section.
1.  This is not a weakness per se, but the category of approach is unsupervised learning on an objective that relaxes the variables to the interval [0,1], always begs the question of how well Adam would have done on the same objective.  To be clear, I don’t mean to train a neural network with Adam, i mean to directly optimize the objective.  It’s a very simple baseline, that is typically very informative, and tells you how the “neural” component of neural combinatorial optimization is actually adding.
2.  I’m not completely sure the approach is exceptionally novel.  Maybe explain how your work relates to “HyperSAT: Unsupervised Hypergraph Neural Networks for Weighted MaxSAT Problems”.  I’m not saying the approaches are the same, far from it, but using a hypergraph GNN for sat like problems seems unbelievably natural and possibly already explored.
3.  In a literature where we’re not out solving WalkSAT a question must be raised about the framing of the contribution.  It’s completely fair to lose to a sat solver.  It’s also fair to lose to WalkSAT which is an unbelievably simple algorithm relative to a hypergraph neural network.  Then what is the key theoretical innovation that deserves publication?  I can see some theoretical merits to the paper, but it would be good for the authors to elaborate on this.
4.  The paper is ridden with typos, at least use a spell checker.

**Questions:**

I listed the questions in the weaknesses section.

---

### Official Review · Reviewer_ZvR6 · 2025-10-31

**Soundness:** 4
**Presentation:** 4
**Contribution:** 3
**Rating:** 6
**Confidence:** 4

**Summary:**

The paper proposes PB-HGNN, an unsupervised hypergraph neural network framework for solving pseudo-Boolean (PUBO) optimization problems which are functions of binary variables expressed as multilinear polynomials. Each polynomial term defines a hyperedge connecting the variables it involves, enabling message passing between variable and term nodes that directly mirrors the structure of the objective. The model learns to minimize a continuous relaxation of the polynomial without supervision and includes a lightweight memory-augmented refinement step for stability. Experiments on random PUBO and MAX-3-SAT benchmarks show that PB-HGNN performs competitively with heuristic and GNN-based QUBO solvers, demonstrating the potential of hypergraph message passing as a general neural approach to high-order combinatorial optimization.

**Strengths:**

The presentation is organized and easy to follow, moving from the formulation and relaxation to the model design and experiments, with enough technical detail for replication.

The hypergraph setup is general and flexible, able to represent a wide range of NP-hard problems (PUBO, MAX-k-SAT, MAX-CUT, etc.) within one unified framework. This is a major strength in comparison to papers which design architectures for specific problems.

The paper extends neural combinatorial optimization beyond the usual quadratic limit of most GNN or Ising-based methods, allowing it to handle higher-order polynomial terms through a hypergraph formulation.

**Weaknesses:**

The paper is weak on the experimental side. The provided experiments do not seem very conclusive.
The authors present the method on 3-SAT but completely ignore architectures such as NeuroSAT which is very similar to the presented method. They claim that the novelty in their method is the bipartite structure of the GNN but this is exactly what NeuroSAT and other architectures are doing (even though the bipartite graph may be a bit different). I believe that more comprehensive experimental evaluation (different baselines, different problems) would improve the paper significantly.

Relevant work:
 https://arxiv.org/pdf/2309.16941
https://arxiv.org/pdf/2504.01173?

**Questions:**

Could you provide comparison to the NeuroSAT architecture, or some other architecture with a bipartite structure?

Did you tried to remove the epoch-level memory to see how well the model performs without it. Also, the idea of using the computed hidden embeddings as an initialization for a next application of a model is present in the recent work under the name deep supervision (https://arxiv.org/abs/2510.04871) but the authors do it withing one batch. This means these embeddings do not need to be retained in a memory after a given batch is processed. I would like to see whether the epoch-level memory has any benefits with respect to this batch level application.

It would be interesting to see how does the performance of a model behave w.r.t. the degree of a given polynomial. For example, one can generate k-SAT problems for different k and then observe how much is the performance degrading with increasing k (as compared to walkSAT for example). Or whether it is possible to train on low degree problems and transfer to higher degree problems. As mentioned in the weakness section, I believe that more experiments would make the paper stronger.

---

### Official Review · Reviewer_G98k · 2025-11-01

**Soundness:** 2
**Presentation:** 2
**Contribution:** 1
**Rating:** 2
**Confidence:** 4

**Summary:**

The paper proposes a specialized hypergraph neural network arhcitecture to solve polynomial binary optimization problems. They do so in a completely unsupervised manner by training the neural network to optimize the multilinear extension of the discrete objective. The method is tested on minimizing random polynomials and on the max 3-SAT problem.

**Strengths:**

- Crafting a specialist architecture for higher order binary polynomials is a reasonable undertaking and there is certainly room in the literature for an empirical contribution along those lines.
- Training without supervision on the multilinear extensions makes sense and has been successful before. So the core approach of the appear is quite reasonable.

**Weaknesses:**

The experiments are inadequate for a contribution of this kind. This is primarily an empirical paper that proposes a specialist architecture. Therefore, for the paper to earn acceptance the experimental evaluation needs to be comprehensive. That includes:
 - Other strong baselines and datasets for the satisfiability problem should be included. SAT offers an excellent evaluation setting for the architecture because different sat problem distributions have different clause sizes and formulae structures, which will be a great stress test.  Consider refs 1,2,3 for example problem instances and additional baselines. More specifically: Neurosat for a supervised baseline, AnyCSP for RL and the solver from 2 for an unsupervised baseline. For datasets, 1 contains plenty of example datasets including k-clique and domset instances that have been converted to CNF SAT. Other examples include sudoku instances in 4 that can be converted to cnf sat and compared against strong baselines in that setting.
- The experiments on SAT provide inadequate details. For example, the baselines you are using that first appeared on Yau et al. also have time cost in the original paper. This is missing here and I believe it's important for us to understand the efficiency of the approach.
 - Certain parameters of the setting for the proposed method and the baselines are not mentioned for SAT. Do you run your model multiple times to convergence on the SAT instances? How many WalkSAT restarts are you performing. For the baselines from Yau et al. I believe you only present WalkSAT with 1 restart and not the 100 restart one. But for table 2 there is no information on the number of restarts.
- Related work is not adequately discussed. The loss used is an expectation and this has been clearly proposed in the literature before (implicitly in the Erdos GNN paper and explicitly in 5).
- I find that the paper does not adequately study and discuss significant components of the pipeline. For example, since the paper is using the expectation as the loss, it can benefit from rounding along the lines of Erdos GNN with the method of conditional expectation. There are also other ways to choose an unsupervised loss for the problem (See 1 and 2). What motivates the use of this specific one here?

Finally, it's clear the paper was somewhat hastily written since there obvious typos and citation mistakes. Overall, I don't think the paper in its current shape can be accepted but it does contain an interesting idea. If executed well with more thorough experiments that are in line with previous work, the paper has the potential to be a valuable contribution in this space.

1. Li, Zhaoyu, Jinpei Guo, and Xujie Si. "G4satbench: Benchmarking and advancing sat solving with graph neural networks." arXiv preprint arXiv:2309.16941 (2023).
2. Ozolins, Emils, et al. "Goal-aware neural SAT solver." 2022 International joint conference on neural networks (IJCNN). IEEE, 2022.
3. Tönshoff, Jan, et al. "One model, any CSP: graph neural networks as fast global search heuristics for constraint satisfaction." arXiv preprint arXiv:2208.10227 (2022).
4. Giannoulis, Panagiotis, Yorgos Pantis, and Christos Tzamos. "Teaching Transformers to Solve Combinatorial Problems through Efficient Trial & Error." arXiv preprint arXiv:2509.22023 (2025).
5. Karalias, Nikolaos, et al. "Neural set function extensions: Learning with discrete functions in high dimensions." Advances in Neural Information Processing Systems 35 (2022): 15338-15352.

**Questions:**

See above

---

### Note · Authors · 2025-11-13

I have read and agree with the venue's withdrawal policy on behalf of myself and my co-authors.